# Evaluation of the Snow Albedo Retrieved from the Snow Kernel Improved the Ross-Roujean BRDF Model

**Anxin Ding [1,2], Ziti Jiao [1,2,\*], Yadong Dong [1,2], Xiaoning Zhang [1,2], Jouni I. Peltoniemi [3], Linlu Mei [4], Jing Guo [1,2], Siyang Yin [1,2], Lei Cui [1,2], Yaxuan Chang [1,2] and Rui Xie [1]**

1   State Key Laboratory of Remote Sensing Science, Jointly Sponsored by Beijing Normal University and Institute of Remote Sensing and Digital Earth of Chinese Academy of Sciences, Beijing 100875, China
2   Beijing Engineering Research Center for Global Land Remote Sensing Products, Institute of Remote Sensing Science and Engineering, Faculty of Geographical Science, Beijing Normal University, Beijing 100875, China
3   Finnish Geospatial Research Institute (FGI) (former Finnish Geodetic Institute), Geodeetinrinne 2, 02330 Masala, Finland
4   Institute of Environmental Physics, University of Bremen, Otto-Hahn-Allee 1, 28359 Bremen, Germany
\*   Correspondence: jiaozt@bnu.edu.cn; Tel.: +86-10-5880-1730

**Abstract:** The original kernel-driven bidirectional reflectance distribution function (BRDF) models were developed based on soil-vegetation systems. To further improve the ability of the models to characterize the snow surface scattering properties, a snow kernel was derived from the asymptotic radiative transfer (ART) model and used in the kernel-driven BRDF model framework. However, there is a need to further evaluate the influence of using this snow kernel to improve the original kernel-driven models in snow albedo retrieval applications. The aim of this study is to perform such an evaluation using a variety of snow BRDF data. The RossThick-Roujean (RTR) model is used as a framework for taking in the new snow kernel (hereafter named the RTS model) since the Roujean geometric-optical (GO) kernel captures a neglectable hotspot effect and represents a more prominent dome-shaped BRDF, especially at a small solar zenith angle (SZA). We obtained the following results: (1) The RTR model has difficulties in reconstructing the snow BRDF shape, especially at large SZAs, which tends to underestimate the reflectance in the forward direction and overestimate reflectance in the backward direction for various data sources. In comparison, the RTS model performs very well in fitting snow BRDF data and shows high accuracy for all data. (2) The RTR model retrieved snow albedos at SZAs = 30°–70° are underestimated by 0.71% and 0.69% in the red and near-infrared (NIR) bands, respectively, compared with the simulation results of the bicontinuous photon tracking (bic-PT) model, which serve as "real" values. However, the albedo retrieved by the RTS model is significantly improved and generally agrees well with the simulation results of the bic-PT model, although the improved model still somewhat underestimates the albedo by 0.01% in the red band and overestimates the albedo by 0.05% in the NIR band, respectively, at SZAs = 30°–70°, which may be negligible. (3) The albedo derived by these two models shows a high correlation ($R^2 > 0.9$) between the field-measured and Polarization and Directionality of the Earth's Reflectances (POLDER) data, especially for the black-sky albedo. However, the albedo derived using the RTR model is significantly underestimated compared with the RTS model. The RTR model underestimates the black-sky albedo (white-sky albedo) retrievals by 0.62% (1.51%) and 0.93% (2.08%) in the red and NIR bands, respectively, for the field-measured data. The shortwave black-sky and white-sky albedos derived using the RTR model for the POLDER data are underestimated by 1.43% and 1.54%, respectively, compared with the RTS model. These results indicate that the snow kernel in the kernel-driven BRDF model frame is more accurate in snow albedo retrievals and has the potential for application in the field of the regional and global energy budget.

**Keywords:** kernel-driven model; bic-PT model; snow kernel; POLDER; BRDF; snow albedo; angular sampling; model evaluation

## 1. Introduction

Snow albedo plays a crucial role in earth-atmosphere systems through its effects on the regional and global energy budget [1–7]. Snow with a high albedo value determines the amount of solar energy absorbed at the surface, which has a powerful positive feedback effect on climate change [1,3,4,6]. As global warming causes snow cover to decrease and exposes a greater amount of snow-free surface, the radiant energy absorbed at the surface increases, which in return further accelerates the global warming effect. Therefore, snow albedo is a fundamental component of the surface energy balance and has a critical effect on the climate system and hydrological studies [8]. The required absolute accuracy of surface albedo is approximately 0.02–0.05 within 5–10 years for climate models [9–12], and Barry recommends that the demanded accuracy of 0.02 for the snow albedo is reasonable [7,11]. The intrinsic reflectance anisotropy of the snow surface must first be considered to retrieve snow albedo with a high retrieval accuracy [13], which can be quantified by the bidirectional reflectance distribution function (BRDF) [14]. Snow is usually more isotropically scattered than other land surfaces at a small solar zenith angle (SZA), but shows strong forward-scattering signatures, especially at a large SZA [15–17]. Therefore, an accurate description of snow forward-scattering properties is essential for snow reflectance models in estimating snow albedo.

The snow reflectance models can generally be categorized as (1) physical models, including simplified radiation transfer and Monte Carlo ray-tracing models; (2) empirical models; and (3) semi-empirical models. The simplified radiation transfer models are built up by simplifying radiative transfer equations, which have specific physical justifications, and these models include the Wiscombe and Warren (WW) model [18], discrete ordinates radiative transfer (DISORT) model [19], and asymptotic radiative transfer (ART) model [20,21]. The Monte Carlo ray-tracing models use a ray-tracing technique and radiative transfer theory to calculate snow scattering properties, which can accurately simulate bidirectional reflectance factors (BRFs) and snow albedo, and these models include the Mishchenko model and bicontinuous photon tracking (bic-PT) model [22,23]. Therefore, the Monte Carlo ray-tracing models tend to provide an effective method for further examining and validating other snow reflectance models. The empirical model is established by empirical statistical descriptions and correlation analyses of field-measured data to characterize snow bidirectional signatures, which have the advantages of simplicity and fast calculation. However, these models do not have physical justifications and are not universally applicable, e.g., the Minnaert model and Walthall model [24,25]. The semi-empirical models are developed by approximating and simplifying physical models, which greatly reduces the complexity of the physical models. Thus, these models maintain a certain physical justification but have the advantage of being easy for wide use, making them highly suitable to generate global operational albedo products [26–28]. Typical examples of these models include the kernel-driven BRDF models and the Rahman-Pinty-Verstraete (RPV) model [29–31].

The original semi-empirical, kernel-driven BRDF models have been extensively utilized in many aspects of remote sensing due to their global applicability and physically-based algorithms [1,13,32–37], such as generating global albedo products. Despite their wide applications, recent studies have indicated that kernel-driven BRDF models that were originally derived based on simplified scenarios for the continuous and discrete vegetation canopies of soil-vegetation systems must be further developed for snow scatterings. To characterize strong forward-scattering behaviours at large illuminating and viewing geometries, Qu et al. added a forward-scattering kernel through the simplification of the RPV model to the kernel-driven BRDF model framework to estimate the surface albedo from multi-angle reflectances of the top of the atmosphere [38]. Recently, Jiao et al. derived a snow kernel from the ART model in the kernel-driven BRDF model framework to better model the anisotropic reflectance of the

snow surface [32], as the RPV model was not specifically developed for investigating snow scattering properties. This snow kernel in the kernel-driven BRDF model framework has been validated by a variety of snow BRDF data and has been reported to have high accuracy in capturing the snow bidirectional signatures. Most recently, an assessment of the performance of these two snow kernels in characterizing snow scattering properties was performed [39]. The new snow kernel improved snow BRDF model has been compared with full radiative transfer simulations performed using SCIATRAN [40], and the results show that the model more accurately characterizes the anisotropic properties of snow, especially in forward directions. However, there is a need to examine the snow kernel performance in the kernel-driven BRDF model framework in retrieving snow albedo compared to the original form of the kernel-driven BRDF model. This examination may provide insight into the potential for this kernel to improve the albedo retrieval accuracy for future applications.

In this study, we perform such an assessment using various data sources. First, we comprehensively evaluate the performance of the latest and original kernel-driven models using 1600 sets of BRFs and albedo simulated by the bic-PT model. Then, we collect 96 sets of field-measured BRFs and 682 sets of the Polarization and Directionality of the Earth's Reflectances (POLDER) BRFs for further evaluation based on the simulated data results. Finally, we discuss the uncertainties in these snow data and summarize the findings of this study.

## 2. Materials

### 2.1. Simulated Data of the bic-PT Model

The model evaluation through the use of a simulation model is extremely important for testing the kernel-driven model performance in estimating snow albedo [41–43]. On the one hand, snow reflectance models can simulate a reference dataset of snow parameters (e.g., equivalent grain radius and snow density) under various conditions. On the other hand, a sufficient amount of multi-angle data with good angular sampling is generally needed to estimate snow albedo, which can be easily controlled by changing the parameters of the snow reflectance models. Therefore, the bic-PT model has been selected to perform this assessment for this purpose. This model was originally proposed by Xiong et al. based on the Monte Carlo ray-tracing technique, which is reported to simulate BRFs and snow albedo more accurately than those simulated with the Mishchenko model [22]. The bic-PT model did not assume specifically shaped particles (e.g., spheres, cubes and cylinders) due to the complexity of the terrain snow microstructure. Additionally, the model considered snow grain shape as a bicontinuous medium, which agreed reasonably well with the real snow microstructure. For the parameter settings listed in Table 1, we referred to related studies [1,17,22,44], and the snow depth was set to 1.5 m to exclude the soil reflectance effect. Note that the albedo simulated by the bic-PT model corresponds to the black-sky albedo because atmospheric effects do not occur in the context of model simulations. Therefore, we simulated 1600 (10 types of equivalent grain radii × 4 types of structure parameters (b) × 5 types of snow densities × 8 types of SZAs) groups of BRFs and albedos in different bands. These simulated albedos were regarded as the "validation" data to assess the performance of the kernel-driven BRDF models in retrieving snow albedo.

**Table 1.** The input parameters of the bicontinuous photon tracking (bic-PT) model.

| Parameters | Value | Step | Unit |
|---|---|---|---|
| Monte Carlo superposition (N) | 1000 | - | - |
| photon total | 50,000 | - | - |
| equivalent grain radius | 0.05–0.50 | 0.05 | mm |
| structure parameter (b) | 1–16 | 5 | - |
| snow density | 0.1–0.5 | 0.1 | g/cm$^3$ |
| wavelength number | 2 | - | - |
| wavelength | 0.67–0.865 | 0.195 | µm |
| solar zenith angle | 0–70 | 10 | degrees (°) |
| view zenith angle | 5–70 | 5 | degrees (°) |
| relative azimuth angle | 1–359 | 2 | degrees (°) |
| snow depth | 1.5 | - | m |
| soil reflectance | 0 | - | - |
| streams number | 32 | - | - |
| order number | 4 | - | - |

## 2.2. Field-Measured Data

To further assess the performance of these two models, we collected 121 sets of field-measured BRFs (http://webdisk.kotisivut.com/fgi/Reflectance_Library/) [45–49], and the details are displayed in Table 2. These BRF data were measured using the Finnish Geodetic Institute's Field Goniospectropolariradiometer (FIGIFIGO), which was an ASD FieldSpec Pro FR Spectroradiometer with a spectral range of 350–2500 nm. The FIGIFIGO is a portable instrument and consists of the goniometer body, turning arm, and laptop computer, which could measure a snow surface sample 20–50 cm in diameter. The view zenith angle (VZA) range was ±80° from nadir, and the zenith angular resolution was approximately 5°. The acquired BRFs had an accuracy of 1–5% depending on wavelength, sample properties, and measurement conditions. The snow sample included new snow, old snow, and dirty snow (e.g., dust, volcanic sand, soot, and silt). In this study, we used the pure snow data with high quality and further screened out the datasets with SZAs > 70° and VZAs > 70°. There were 96 sets of BRFs selected for the model assessment.

**Table 2.** The location, targets and measurement dates of the field-measured bidirectional reflectance factors (BRFs).

| Date | Site | Latitude | Longitude | Sample |
|---|---|---|---|---|
| Apr. 2005 | Sodankylä, Etupiha | 67.0021° | 27.2430° | natural snow |
| Apr. 2007 | Tahtela, Sodankylä | 67.3622° | 26.6344° | natural snow |
| Apr. 2008 | Sodankylä | 67.3628° | 26.6355° | new snow; old snow |
| Mar. 2009 | Masala | 60.1719° | 24.5542° | natural snow |
| Apr. 2009 | Kommattivaara, Sodankylä | 67.4211° | 26.7923° | natural snow |
| Jun.–Jul. 2010 | Summit | 72.5961° | -38.4219° | natural snow |
| Mar. 2010 | Sodankylä | 67.3627° | 26.6356° | natural snow |
| Mar. 2013 | Luoman Asema | 60.1721° | 24.5486° | natural snow; snow + dust |
| Apr. 2013 | Sodankylä | 67.3958° | 26.6141° | natural snow; snow + volcanic sand, soot, and silt |

## 2.3. POLDER BRDF Database

The POLDER BRDF database was described in detail in previous studies [1,15,32,34,35,50] and the database had a spatial resolution of approximately 6 × 7 km. The POLDER radiometer onboard

the Polarization & Anisotropy of Reflectances for Atmospheric Sciences coupled with Observations from Lidar (PARASOL) satellite acquired data in 9 spectral channels, in which 3 of which (490, 670 and 865 nm) measured the polarized reflectance of the incident light. During the PARASOL satellite overpass, the surface targets can be consecutively viewed up to 16 times. Therefore, a major specificity of this POLDER radiometer is its multidirectional capability. For the purpose of this study, we also selected pure snow data. Additionally, we removed the datasets with SZAs > 70° and VZAs > 70° [32]. Finally, 682 pixels were selected for model comparison. We obtained the statistics of the SZA distributions for these datasets. The results are shown in Figure 1. The SZA distribution of the POLDER and field-measured data are mainly in the range of 50°–70° with proportions of 85.9% and 80.3%, respectively. In addition, the shortwave albedo was generated by the narrowband to broadband coefficients for the POLDER data [51,52], and the conversion formula can be expressed as follows:

$$\alpha_{POLDER} = 0.112\alpha_1 + 0.388\alpha_2 - 0.266\alpha_3 + 0.668\alpha_4 + 0.0019 \tag{1}$$

where $\alpha_{POLDER}$ is the shortwave albedo, and $\alpha_i$ represents the spectral albedos.

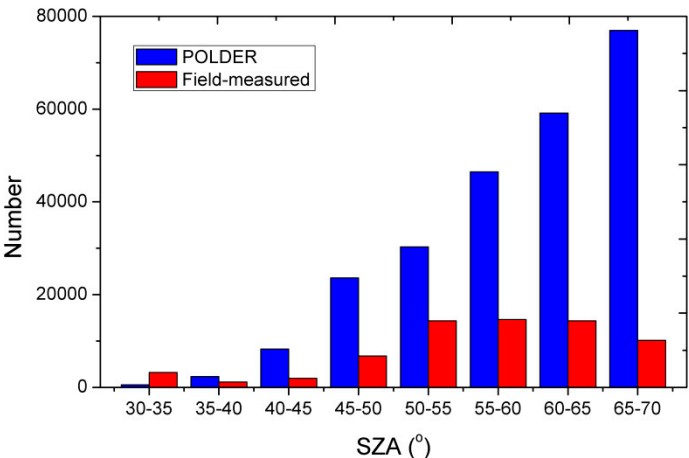

**Figure 1.** The solar zenith angle (SZA) distribution of the Polarization and Directionality of the Earth's Reflectances (POLDER) and field-measured data.

## 3. Models and Methods

### 3.1. Kernel-Driven Model

The semi-empirical kernel-driven model was originally proposed by Roujean et al. [31], which considered that surface reflectance is the sum of three components, and the equation can be expressed as follows:

$$R(\theta_s, \theta_v, \varphi, \lambda) = f_{iso}(\lambda) + f_{vol}(\lambda)K_{vol}(\theta_s, \theta_v, \varphi) + f_{geo}(\lambda)K_{geo}(\theta_s, \theta_v, \varphi). \tag{2}$$

where $R(\theta_s, \theta_v, \varphi, \lambda)$ is the surface reflectance in waveband $\lambda$, which is a function of the VZA $\theta_v$, SZA $\theta_s$ and relative azimuth angle (RAA) $\varphi$. The $f_{iso}(\lambda)$, $f_{vol}(\lambda)$ and $f_{geo}(\lambda)$ are the weight components of the isotropic scattering kernel, volume scattering kernel $K_{vol}(\theta_s, \theta_v, \varphi)$ and geometric-optical (GO) scattering kernel $K_{geo}(\theta_s, \theta_v, \varphi)$, respectively.

First, we retrieve the three optimal weight parameters (i.e., $f_{iso}(\lambda)$, $f_{vol}(\lambda)$ and $f_{geo}(\lambda)$) based on the least-squares method and they are constrained as non-negative values. Then, the black-sky albedo (BSA) and white-sky albedo (WSA) are retrieved by weighting the kernel's integral with the model

parameters for these two kernel-driven models used for the albedo comparison. The equations of BSA and WSA can be expressed as follows:

$$\text{BSA}(\theta_s, \lambda) = \frac{1}{\pi} \int_0^{2\pi} \int_0^{\pi/2} R(\theta_s, \theta_v, \varphi, \lambda) sin\theta_v cos\theta_v d\theta_v d\varphi \tag{3}$$

$$\text{WSA}(\lambda) = 2 \int_0^{\pi/2} \text{BSA}(\theta_s, \lambda) sin\theta_s cos\theta_s d\theta_s \tag{4}$$

The actual albedo is influenced by the diffuse and direct irradiance under ambient illumination conditions, which is a linear combination of BSA and WSA. The equation of actual albedo can be expressed as follows:

$$\alpha(\theta_s) = [1 - S]\text{BSA}(\theta_s) + S \cdot \text{WSA} \tag{5}$$

where S refers to the proportion of diffuse light, which is a function of aerosol optical depth, aerosol type and wavelength, and $\alpha(\theta_s)$ refers to the actual albedo.

### 3.1.1. RTR Model

In this study, we adopt a combination of the RossThick and Roujean kernels (hereafter referred to as the RTR model) because the RTR model characterizes a prominent dome-shaped BRDF curve with a low hotspot signature, which is consistent with snow directional scattering, especially at a small solar zenith angle. The RossThick kernel was proposed by Roujean based on the radiative transfer theory [31,33,50,53], which describes the anisotropic scattering of horizontal homogeneous vegetation. The Roujean kernel assumes a surface placed on a flat horizontal plane [31,53], which contains a large number of vertical opaque protrusions. The equations of the $K_{vol}$ and $K_{geo}$ kernels can be expressed as follows:

$$K_{vol} = \frac{\left(\frac{\pi}{2} - \xi\right)\cos\xi + \sin\xi}{\cos\theta_s + \cos\theta_v} - \frac{\pi}{4} \tag{6}$$

$$cos\xi = \cos\theta_s \cos\theta_v + \sin\theta_s \sin\theta_v \cos\varphi \tag{7}$$

$$K_{geo} = \frac{1}{2\pi}[(\pi - \varphi)\cos\varphi + \sin\varphi]\tan\theta_s \tan\theta_v - \frac{1}{\pi}[\tan\theta_s + \tan\theta_v + K] \tag{8}$$

$$K = \sqrt{\tan^2\theta_s + \tan^2\theta_v - 2\tan\theta_s \tan\theta_v \cos\varphi} \tag{9}$$

where $\xi$ is the phase angle and $K$ describes the horizontal distance between the illuminating and viewing directions [54].

### 3.1.2. RTS Model

Recently, Jiao et al. proposed a snow kernel in the kernel-driven BRDF model framework to consider snow scattering characteristics [32]. Thus, we further improve the RTR model by replacing the GO scattering kernel with the snow kernel in Equation (2) to characterize snow scatterings (hereafter referred to as the RTS model). The snow kernel equations can be expressed as follows:

$$K_{snw}(\theta_s, \theta_v, \varphi) = R_0(\theta_s, \theta_v, \varphi)(1 - \alpha f(\theta_s, \theta_v, \varphi)) + 0.4076\alpha - 1.1081 \tag{10}$$

$$R_0(\theta_s, \theta_v, \varphi) = \frac{C_1 + C_2(\cos\theta_s + \cos\theta_v) + C_3 \cos\theta_s \cos\theta_v + P(\xi)}{4(\cos\theta_s + \cos\theta_v)} \tag{11}$$

$$P(\xi) = 11.1e^{-0.087(180-\xi)} + 1.1e^{-0.014(180-\xi)} \tag{12}$$

$$f(\theta_s, \theta_v, \varphi) = \cos\xi \cdot e^{-\cos\xi} \tag{13}$$

where $K_{snw}(\theta_s, \theta_v, \varphi)$ and $f_{snw}(\lambda)$ are the snow kernel and its weight components, respectively. $R_0(\theta_s, \theta_v, \varphi)$ is the surface reflectance of the snow at zero absorption, and the values of $C_1$, $C_2$

and $C_3$ are 1.247, 1.186 and 5.157, respectively. Note that $\xi$ in Equation (12) represents the phase angle in degrees. $f(\theta_s,\theta_v,\varphi)$ is the correction component of the ART model as a function of phase angle $\xi$. The $\alpha$ value of 0.3 was derived by fitting the global POLDER BRDF database. In this study, we use the optimal $\alpha$ value for the specific snow BRDF data on a case-by-case basis because we have sufficient observations in the principal plane (PP) to determine the optimal $\alpha$ value.

### 3.2. Evaluation Method for the Kernel-Driven Models.

In this section, we introduce the procedure of model evaluation. First, we make a comprehensive assessment of these two kernel-driven models using simulation data, including the characterization of snow BRDF signatures, retrieval of snow albedo, and investigation of the effects of different angular samples on retrieving snow albedo. Then, we further compare the difference between these two models in retrieving albedo using field-measured and satellite BRF data. In this paper, the root mean square error (RMSE), bias and relative error (RE) values are applied as the quality assessment indices, and the T-test is used to test the differences in these two models in retrieving snow albedo. The RMSE, bias, RE and T-test equations can be expressed as follows:

$$\text{RMSE} = \sqrt{\frac{\sum_{i=1}^{n}(A_1 - A_2)^2}{n-3}} \tag{14}$$

$$\text{bias} = \frac{\sum_{i=1}^{n}(A_1 - A_2)}{n} \tag{15}$$

$$\text{RE} = \sqrt{\frac{\sum_{i=1}^{n}(\text{ABS}(A_1 - A_2)/A_2)}{n}} \tag{16}$$

$$T = \frac{|X_1 - X_2|}{\sqrt{\frac{(N_1-1)S_1^2 + (N_2-1)S_2^2}{N_1+N_2-2}\left(\frac{1}{N_1} + \frac{1}{N_2}\right)}} \tag{17}$$

where $A_1$ represents the snow reflectance or albedo retrieved by these two kernel-driven models, and $A_2$ represents the snow reflectance or albedo simulated by the bic-PT model. $X_1$ and $X_2$ are the sample means, $S_1$ and $S_2$ are the sample standard deviations, and $N_1$ and $N_2$ are the numbers of samples.

## 4. Results and Analysis

### 4.1. Results Using the bic-PT Model

#### 4.1.1. Evaluating the Models in Fitting the Snow Simulation BRDFs

We first generate a set of typical simulation snow BRDF data using the bic-PT model as a case study. The equivalent grain radius is 0.1 mm, the b parameter is 1, and the snow density is 0.1 g/cm$^3$. This aims to display the difference of these two kernel-driven models in fitting snow BRDF simulation data at three typical SZAs (i.e., SZAs = 0°, 40° and 70°) by the bic-PT model. Figure 2 shows the predicted reflectance using these two models and the simulated reflectance for the three typical SZAs in the PP in the red band (670 nm). The BRDF shape reconstructed by the RTR model shows a large inconsistency with the simulation reflectances as a function of the SZAs, but the RTS model fits the simulated data very well. The BRDF shape reconstructed by the RTR model at SZA = 0° agrees reasonably well with the simulated data, particularly at VZA < 70°. At VZA > 70°, the RTR model tends to underestimate the observations relative to the RTS model. However, at SZA = 40°, the result of the RTR model tends to underestimate (overestimate) reflectance in the forward (backward) direction. When SZA = 70°, the RTR model does not significantly characterize strong scattering of snow in the forward direction, which however is well captured by using the snow kernel of RTS model. Table 3 shows the BRDF parameters and statistics for these two models. It is clear that the determination

coefficient ($R^2$) derived by the RTR model has a large variation in the range of 0.33–0.99, depending on the SZA values. In contrast, the RTS model is highly consistent with the simulated data at different SZAs, and the $R^2$ ranges from 0.94–1.00, indicating that the RTS model fits these simulated data more accurately than does the RTR model.

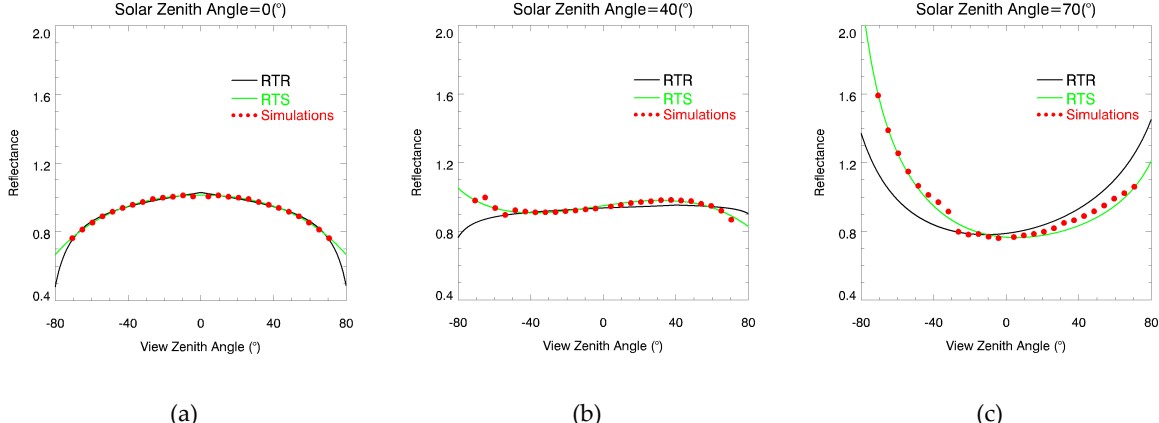

(a)  (b)  (c)

**Figure 2.** The simulated data of the bicontinuous photon tracking (bic-PT) model (red dots) and the reconstructed bidirectional reflectance distribution function (BRDF) shapes by the RossThick-Snow (RTS) model (green) and RossThick-Roujean (RTR) model (black) in the principal plane (PP) of the red band (670 nm), where (**a**), (**b**) and (**c**) represent the SZAs equal to 0°, 40° and 70°, respectively. The equivalent grain radius is 0.1 mm, the b parameter is 1, and the snow density is 0.1 g/cm³.

**Table 3.** The reconstructed bidirectional reflectance distribution function (BRDF) parameters and statistics of these two kernel-driven models for three typical solar zenith angles (SZAs) (i.e., SZA = 0°, 40° and 70°), and the BRDF parameter $f_{\text{geo}}$ for the RossThick-Snow (RTS)model is $f_{\text{snw.}}$ The $f_{\text{iso}}(\lambda)$, $f_{\text{vol}}(\lambda)$, $f_{\text{geo}}(\lambda)$ and $f_{\text{snw}}(\lambda)$ are the weight components of the isotropic scattering, volume scattering, geometric-optical scattering and snow kernels, respectively.

| Model | SZA (°) | $f_{\text{iso}}$ | $f_{\text{vol}}$ | $f_{\text{geo}}$ | $R^2$ | RMSE | Bias | $\alpha$ |
|---|---|---|---|---|---|---|---|---|
|  | 0 | 1.026 | 0.020 | 0.151 | 0.991 | 0.007 | 0.000 | – |
| RTR | 40 | 0.961 | 0.000 | 0.048 | 0.333 | 0.031 | 0.000 | – |
|  | 70 | 0.786 | 0.301 | 0.000 | 0.495 | 0.105 | 0.000 | – |
|  | 0 | 0.868 | 0.000 | 1.158 | 0.999 | 0.003 | 0.000 | 0.00 |
| RTS | 40 | 0.869 | 0.411 | 1.960 | 0.936 | 0.009 | 0.000 | 0.05 |
|  | 70 | 0.845 | 0.167 | 0.538 | 0.965 | 0.028 | 0.000 | 0.30 |

We further examine and compare the performance of these two models in fitting 200 sets of simulation data in the red and near-infrared (NIR) bands. Figure 3 shows the scatterplots between the reflectance retrieved by these two models and the BRF data simulated by the bic-PT model. The overall $R^2$ value derived by the RTS model ranges from 0.88–0.93, and the RMSE results are approximately 0.02. In contrast, the overall $R^2$ value of the RTR model is much smaller than that of the RTS model at approximately 0.21 (0.56) in the red (NIR) bands, and the RMSE derived by the RTR model is approximately 0.06 in both bands. In general, these results show that the RTS model captures a higher accuracy than the RTR model in fitting these simulated data in the red and NIR bands. Figure 4 also shows that the light-coloured part of the BRF data simulated by the bic-PT model varies largely, these data may lead to some uncertainties for the evaluation of these two models. However, most BRF data have high consistency and any uncertainty mainly results from a few simulated data.

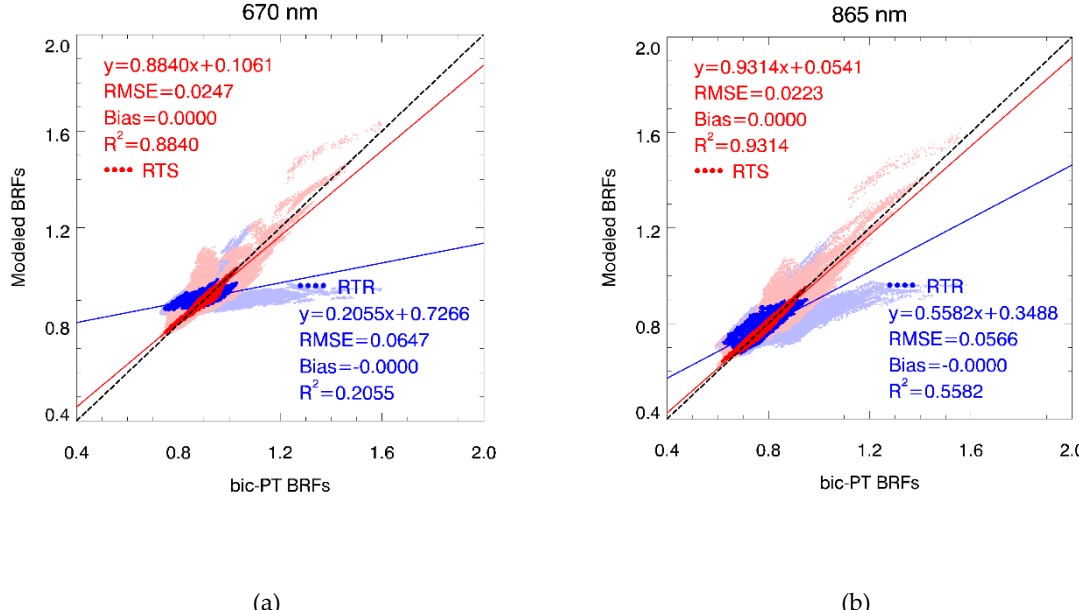

(a)　　　　　　　　　　　　　　　　　　　　　(b)

**Figure 3.** The density plots comparing the simulated data with the RossThick-Roujean (RTR)model (blue dots) and RossThick-Snow (RTS)model (red dots) in the 670 nm (**a**) and 865 nm (**b**) bands.

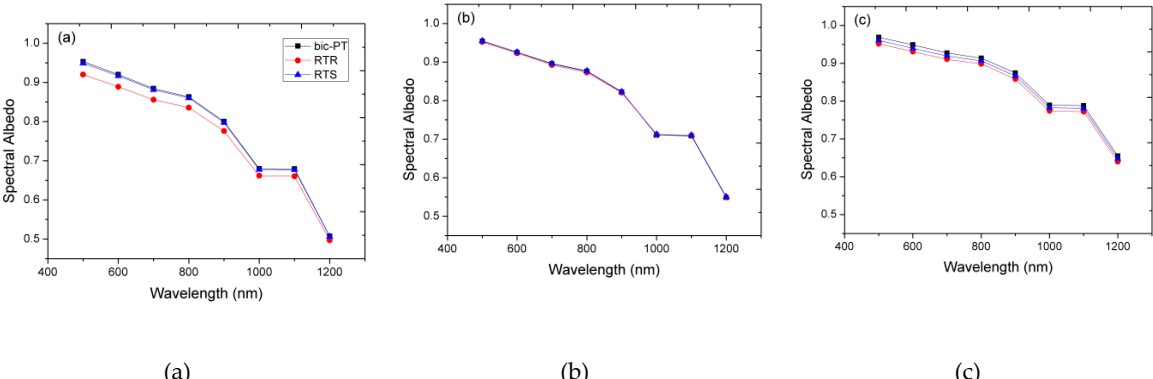

(a)　　　　　　　　　　　　　(b)　　　　　　　　　　　　　(c)

**Figure 4.** The spectral albedo simulated by the bic-PT model (black) and the spectral albedo retrieved by the RTS (blue) and RTR models (red), where (**a**), (**b**) and (**c**) represent SZAs of 0°, 40° and 70°, respectively. The equivalent grain radius is 0.1 mm, the b parameter is 1, and the snow density is 0.1 g/cm$^3$.

### 4.1.2. Evaluating the Models in Estimating Snow Albedo

In this section, we analyse the difference between the two models for the estimation of snow spectral albedo through a comparison with the bic-PT model using a set of simulation data that was already used and showed as a case study in Figure 2. First, we select a spectral range of 500–1200 nm at 100 nm intervals, and the results of the spectral albedo at SZA = 0°, 40° and 70° are shown in Figure 4. The spectral albedo retrieved by these two models presents a general consistency with the simulation results by the bic-PT model at these SZAs because the albedos are the integrals of the BRDF and the lack of consistency of the BRDF shapes between models does not necessarily have a large influence on the albedo retrieval. Even so, the spectral albedo retrieved by the RTR model somewhat varies with the SZAs. Although the BRDF shape at SZA = 0° reconstructed by the RTR model is highly consistent with the bic-PT model at VZA < 70° (Figure 2a), the spectral albedo retrieved by this model is significantly underestimated, most probably due to the underestimation of observations at VZA > 70° by the RTR model. Although the RTR model at SZA = 40° does not perform well in characterizing snow BRDF shapes (Figure 2b), the spectral albedo retrieval is nearly perfectly consistent with the simulation result by the bic-PT model. The reason is most probably due to the overestimation of this model in

the backward direction which is significantly compensated by the underestimation in the forward direction in the integrals of the BRDF shapes. At SZA = 70°, the spectral albedo retrieved by the RTR model is in general agreement with the bic-PT model, but the RTR model somewhat underestimates the simulated results because snow tends to have high reflectance values in the forward-scattering direction at SZA = 70°, which is not captured by the RTR model at all. In contrast, the spectral albedo retrieved by the RTS model captures a nearly perfect consistency with the simulated spectral albedo, at three SZAs in all bands although the RTS model result at SZA = 70° shows a very small deviation from the simulation results of the bic-PT model (we will test the significance of such deviation later on). In general, the spectral albedo retrieved by the RTR model shows more inconsistencies with the simulation results of the bic-PT model as a function of the SZA, somewhat depending on the different wavelengths. The spectral albedo retrieved by the RTR model has a larger difference in the visible bands than the NIR bands. However, the RTS model presents an improved result relative to the RTR model in retrieving snow spectral albedo.

Then, we focus on performing a similar assessment using 1600 groups of the entire simulation BRFs and albedo generated by the bic-PT model at two wavelengths centred at 670 nm and 865 nm. The results are shown in Figure 5. Obviously, these two models present a high correlation ($R^2$ > 0.88) with the simulation albedo, and the RMSE results are less than 0.016 in the red and NIR bands. However, the snow albedo retrieved by the RTR model seems to somewhat underestimate these "validation" albedos. The RTR model result underestimates 1.39% and 1.21% in the red and NIR bands, respectively, which has a statistically significant difference from the "validation" albedo (P < 0.05). In addition, the albedo retrieved by the RTR model is partly out of the range of ±0.02, arriving at 28.00% and 7.88% in the red and NIR bands, respectively. However, the snow albedo retrieved by the RTS model agrees reasonably well with the "validation" albedo (P > 0.05) and all the points are distributed along the 1:1 line and fall within the range of ±0.02, which shows an insignificant underestimation of less than 0.14% and 0.08% in the red and NIR bands, respectively. Clearly, the RTS model performs significantly better than the RTR model in retrieving snow albedo compared with these "validation" albedos.

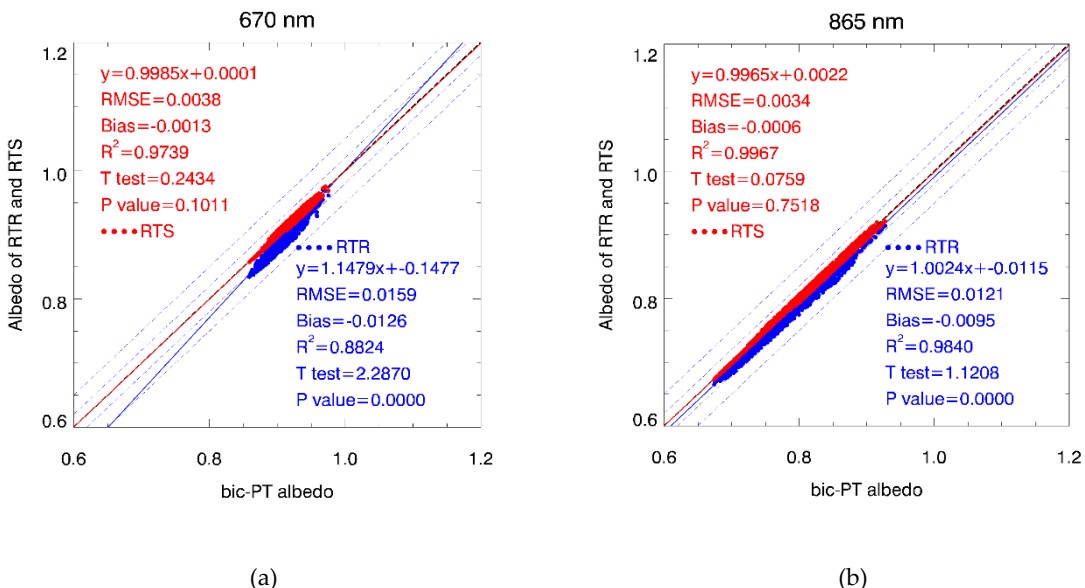

(a)　　　　　　　　　　　　　　　　　　　　　(b)

**Figure 5.** The comparison of albedos retrieved by these two kernel-driven models with the albedo simulated by the bic-PT model, and the two wavelengths are 670 nm (**a**) and 865 nm (**b**). The central dotted lines are 1:1 lines, the outer dotted lines are 0.02 offset lines, and the outermost dotted lines are 0.05 offset lines.

Since snow is mainly distributed in the middle and high latitudes, the SZA is generally large (Figure 1). Therefore, we only use data with SZAs = 30°–70° to compare the snow albedo retrieved by

these two kernel-driven models with the "validation" data. The results are shown in Figure 6. The snow albedo retrieved by the RTS model matches the bic-PT model simulation albedo very well with $R^2$ at greater than 0.95, although the albedos retrieved by the RTS model at SZAs = 30°–70° are somewhat underestimated by < 0.01% in the red band and overestimated by < 0.05% in the NIR band, which is most probably due to the uncertainties by the bic-PT model. In comparison, the albedo retrieved by the RTR model shows a statistically significant difference from the "validation" data in the red and NIR bands (P <0.05). The albedo retrieved by the RTR model is significantly underestimated by 0.71% and 0.69% in the red and NIR bands, respectively. The albedo retrieved by these two kernel-driven models lies around the 1:1 lines and mainly falls within the range of ±0.02. In general, these results demonstrate that the RTS model presents an improved ability in retrieving snow albedo compared with the RTR model.

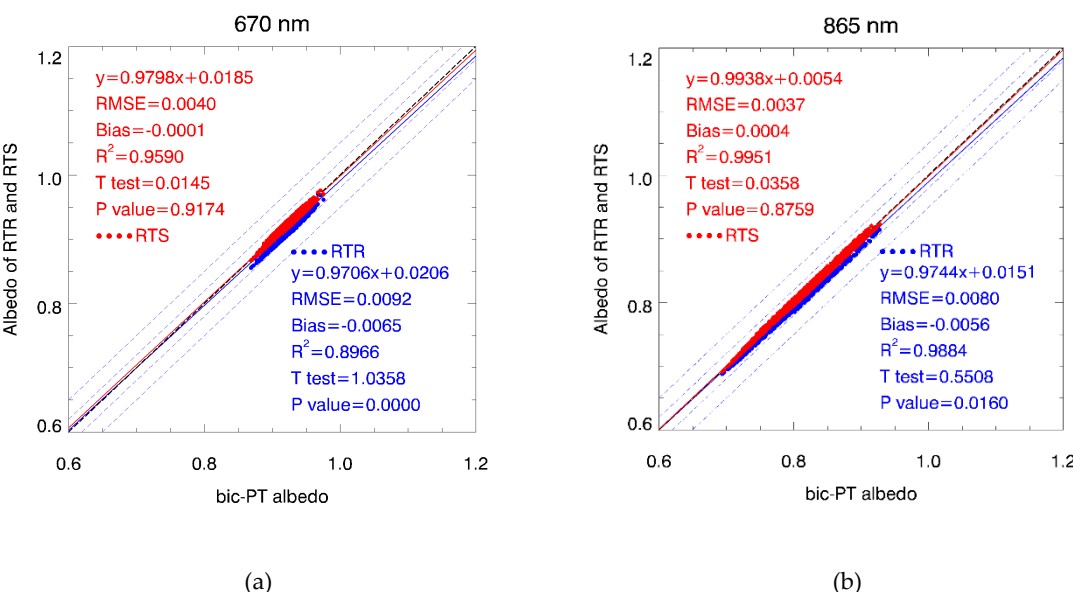

(a)  (b)

**Figure 6.** The comparison of the albedos retrieved by the two kernel-driven models with the "validation" albedo, and the two wavelengths are 670 nm (**a**) and 865 nm (**b**). The central dotted lines are 1:1 lines, the outer dotted lines are 0.02 offset lines, and the outermost dotted lines are 0.05 offset lines.

### 4.1.3. Investigating the Angular Sampling Influence on Snow Albedo Retrieval

To estimate the snow albedo from remote sensing observations, a sufficient amount of multi-angle data with a good angular sampling is generally required [55,56]. However, different off-nadir satellite sensors are designed with different sampling strategies. Thus, the satellites can only obtain specifically distributed observations in a given location over a given period, e.g., the Multi-angle Imaging Spectroradiometer (MISR) and the Moderate Resolution Imaging Spectroradiometer (MODIS) [13,57]. Therefore, we investigate the effects of different angular samplings on retrieving snow albedo for these two models using 1600 sets of simulated BRFs. First, we select 10 different angle samplings according to the distance from the PP to the cross principal plane (CPP) in the range of 1°–91° with an internal angle of 10° (hereafter named ID = 0–9). Figure 7 shows the angle distribution of the simulated data by the bic-PT model and three patterns of typical angular samplings (i.e., ID = 0, 4 and 9), and the simulation data with the entire spatial samplings are regarded as the benchmarked data to validate the results derived by using the azimuthal sampling BRDF data.

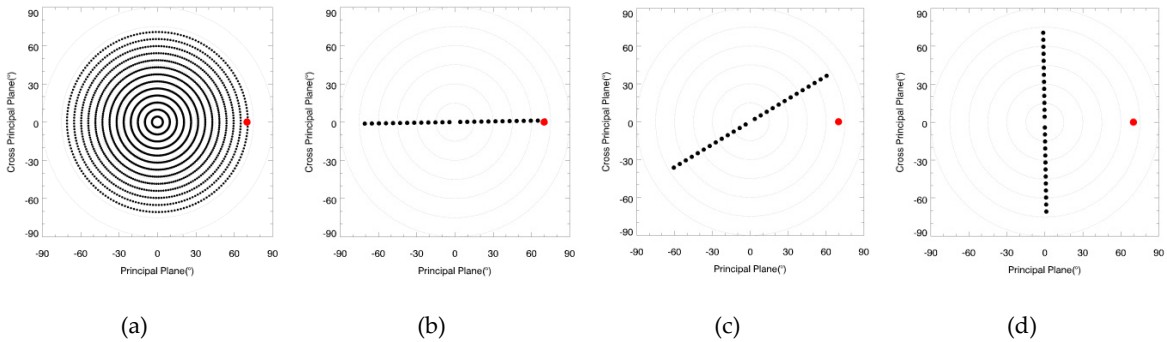

**Figure 7.** The angle distribution of the "validation" data (**a**) and three typical azimuthal samplings (**b**), (**c**) and (**d**) (i.e., ID = 0, 4 and 9), where the red dots represent the sun's direction, and the black dots represent the view directions.

Tables 4 and 5 show the statistical results of these two kernel-driven models for the different angular samplings in the red band. The albedo provided by these two kernel-driven models shows high correlations with the "validation" data, especially when the ID ≤ 6. As the ID increases (i.e., the observation angle moves from PP to CPP), the albedo provided by these two models is first overestimated and then underestimated. This change may be because the snow shows a strong anisotropic reflectance in the PP but shows more isotropic reflectance in the CPP. The albedo provided by these two kernel-driven models shows higher accuracy compared with "validation" data when ID = 4 and 5. However, when ID > 6, the albedos provided by these two kernel-driven models are significantly underestimated because these data provide less BRDF information, which cannot be used to reconstruct the snow BRDF shape accurately, which significantly affects the albedo retrieval, particularly in the exact CPP. The average REs of the RTS model are much smaller than those of the RTR model, which indicates that the RTS model shows greater stability and higher accuracy for different angular samplings, while the RTR model shows more uncertainty.

**Table 4.** The statistical results of the RossThick-Roujean (RTR) model for the different angular samplings in the red (670 nm) band.

| ID | $R^2$ | RMSE | Bias | RE(%) | T-test | P value |
|----|-------|------|------|-------|--------|---------|
| 0 | 0.929 | 0.018 | 0.014 | 1.462 | 2.177 | 0.000 |
| 1 | 0.937 | 0.017 | 0.013 | 1.387 | 2.080 | 0.000 |
| 2 | 0.956 | 0.015 | 0.011 | 1.184 | 1.798 | 0.000 |
| 3 | 0.976 | 0.010 | 0.008 | 0.836 | 1.283 | 0.000 |
| 4 | 0.982 | 0.006 | 0.003 | 0.406 | 0.546 | 0.001 |
| 5 | 0.968 | 0.005 | -0.003 | 0.439 | 0.432 | 0.008 |
| 6 | 0.896 | 0.013 | -0.009 | 1.052 | 1.690 | 0.000 |
| 7 | 0.673 | 0.023 | -0.016 | 1.847 | 3.042 | 0.000 |
| 8 | 0.377 | 0.031 | -0.022 | 2.493 | 4.076 | 0.000 |
| 9 | 0.495 | 0.026 | -0.018 | 2.039 | 3.257 | 0.000 |

**Table 5.** The statistical results of the RossThick-Snow (RTS) model for the different angular samplings in the red (670 nm) band.

| ID | $R^2$ | RMSE | Bias | RE(%) | T-test | P value |
|----|-------|------|------|-------|--------|---------|
| 0 | 0.877 | 0.008 | 0.003 | 0.602 | 0.493 | 0.001 |
| 1 | 0.894 | 0.008 | 0.003 | 0.551 | 0.495 | 0.001 |
| 2 | 0.925 | 0.007 | 0.003 | 0.451 | 0.507 | 0.001 |
| 3 | 0.952 | 0.005 | 0.003 | 0.364 | 0.478 | 0.001 |
| 4 | 0.976 | 0.004 | 0.002 | 0.293 | 0.337 | 0.021 |
| 5 | 0.991 | 0.002 | 0.000 | 0.175 | 0.005 | 0.971 |
| 6 | 0.952 | 0.007 | -0.003 | 0.410 | 0.576 | 0.000 |
| 7 | 0.838 | 0.015 | -0.008 | 0.972 | 1.397 | 0.000 |
| 8 | 0.633 | 0.025 | -0.016 | 1.904 | 2.824 | 0.000 |
| 9 | 0.599 | 0.030 | -0.020 | 2.391 | 3.429 | 0.000 |

## 4.2. Results with Field-Measured Data

Similarly, we first evaluate the performance of these two kernel-driven models in characterizing snow scattering signatures using the field-measured data. Figure 8 shows the predicted reflectance using these two models and the field-measured reflectance for two typical SZAs (i.e., 50° and 70°) in the PP. Obviously, the RTR model results at SZAs = 50° and 70° tend to underestimate (overestimate) the reflectance in the forward (backward) direction. In contrast, the inversion results of the RTS model are highly consistent with the field-measured data. Table 6 shows the results of the retrieved BRDF parameters and statistics for these two models. The RTR model seems to have difficulty characterizing the snow BRDF of these field-measured data, whereas the RTS model captures high correlations with the field-measured data. The $R^2$ ranges from 0.95–0.96, which indicates that the RTS model performs well in fitting the snow BRDF.

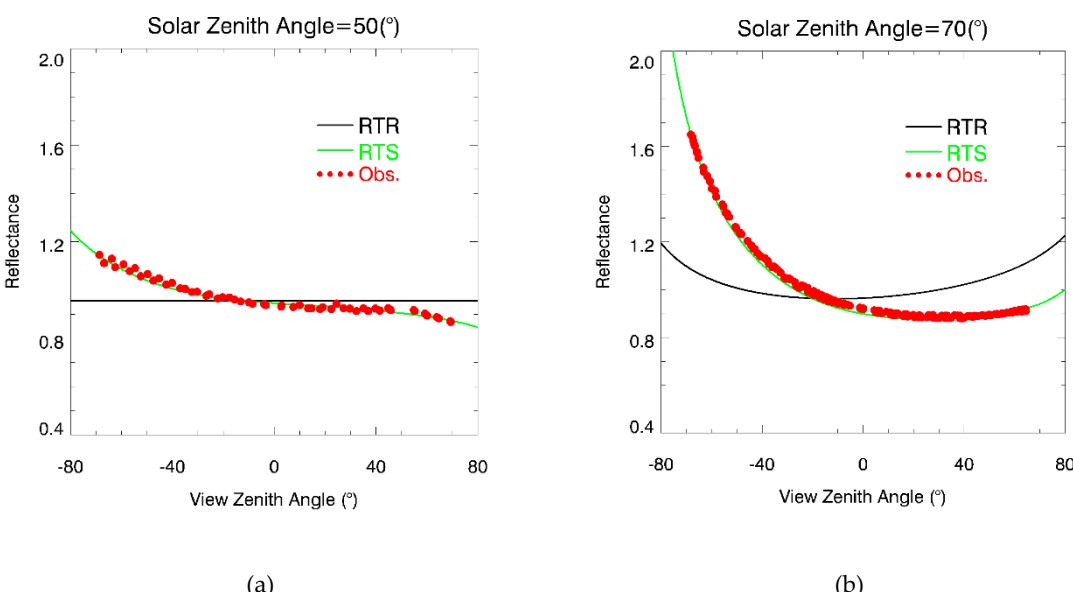

(a)                                                                      (b)

**Figure 8.** The field-measured data (red dots) and reconstructed BRDF shapes using the RTS model (green) and RTR model (black) in the PP and red band (670 nm), where (**a**) and (**b**) represent the cases, and the SZAs equal 50° and 70°, respectively.

**Table 6.** The BRDF parameters and statistics of these two models for the field-measured data, where the BRDF parameter $f_{geo}$ for the RTS model is $f_{snw}$. The $f_{iso}(\lambda)$, $f_{vol}(\lambda)$, $f_{geo}(\lambda)$ and $f_{snw}(\lambda)$ are the weight components of the isotropic scattering, volume scattering, geometric-optical scattering and snow kernels, respectively.

| Model | SZA (°) | $f_{iso}$ | $f_{vol}$ | $f_{geo}$ | $R^2$ | RMSE | Bias | $\alpha$ |
|---|---|---|---|---|---|---|---|---|
| RTR | 50 | 0.954 | 0.000 | 0.000 | 0.000 | 0.050 | 0.000 | – |
| | 70 | 0.964 | 0.119 | 0.000 | 0.056 | 0.161 | 0.000 | – |
| RTS | 50 | 0.973 | 0.000 | 0.838 | 0.950 | 0.011 | 0.000 | 0.19 |
| | 70 | 1.006 | 0.000 | 0.721 | 0.961 | 0.033 | 0.000 | 0.30 |

Then, we assess these two kernel-driven models based on 96 sets of field-measured data in the red and NIR bands as shown in Figure 9. The overall $R^2$ value derived by the RTS model ranges from 0.90–0.92, and the RMSE results are approximately 0.05. In contrast, the RTR model acquires an overall $R^2$ value within the range of 0.58–0.70 with RMSEs at approximately 0.10. Obviously, these results indicate that the RTS model has a higher accuracy than the RTR model in fitting these field-measured data in both the red and NIR bands.

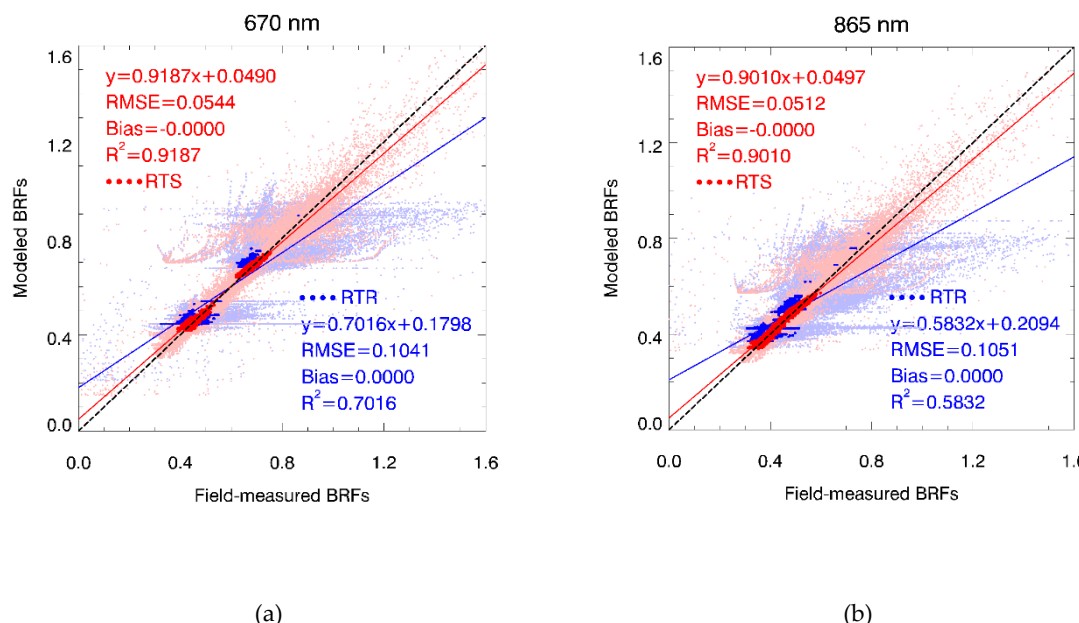

(a)　　　　　　　　　　　　　　　　　　　　　　　　(b)

**Figure 9.** The density plots comparing simulated data with the RTR model (blue dots) and RTS model (red dots) in the 670 nm (**a**) and 865 nm (**b**) bands.

Next, we assess and compare the difference between these two kernel-driven models in retrieving snow albedo using 96 sets of BRFs data of field snow measurements. Figure 10 shows the scatter plots that compare the BSA and WSA retrieved by these two models in the red and NIR bands. The BSA is calculated using the average SZA of each set of field measurements that vary in a generally small range of SZA. The retrieved albedos by using these two models are very highly correlated ($R^2$ = ~0.99), especially in the BSA. Their RMSEs are approximately 0.01 (0.02) for the BSA (WSA) in the red and NIR bands. In general, the retrieved albedo by the RTR model is somewhat underestimated relative to the result by the RTS model, arriving at 0.62% (1.51%) and 0.93% (2.08%) for the BSA (WSA) in the red and NIR bands, respectively. Somewhat surprisingly, such differences between the two models are not statistically significant according to the T-test results because the SZAs of these snowfield measurements are within a range of 35°–55°, where the BRDF shapes are relatively flat (Figure 8a). In such a case, although the RTR model cannot fit these measurements well relative to the RTS model, the integral of the modelled BRDFs using the respective models cannot result in significant differences in the retrieved albedos that are actually

dependent on SZA variations, particularly for the BSAs. In addition, a few points are out of the range of ±0.05 in the red and NIR bands. We examine the BRDF shapes of these specific points in Figure 11, which shows the BRDF shapes reconstructed by these two models in the PP of the red band (670 nm). Clearly, the BRDF shapes reconstructed by the RTS model are highly consistent with these field-measured data; however, the RTR model results at a large SZA (~70°) that tends to underestimate the reflectance in the forward direction, resulting in the underestimation of snow albedo retrieval. In general, the retrieved albedos by the RTS model capture more variations than that of the RTR model because the former is able to capture more details of the BRDF variabilities for each set of measurements, particularly in the forward scattering direction. Notably, the uncertainty in the field measurements of snow surface is very obvious in the forward scattering in the PP (Figure 11).

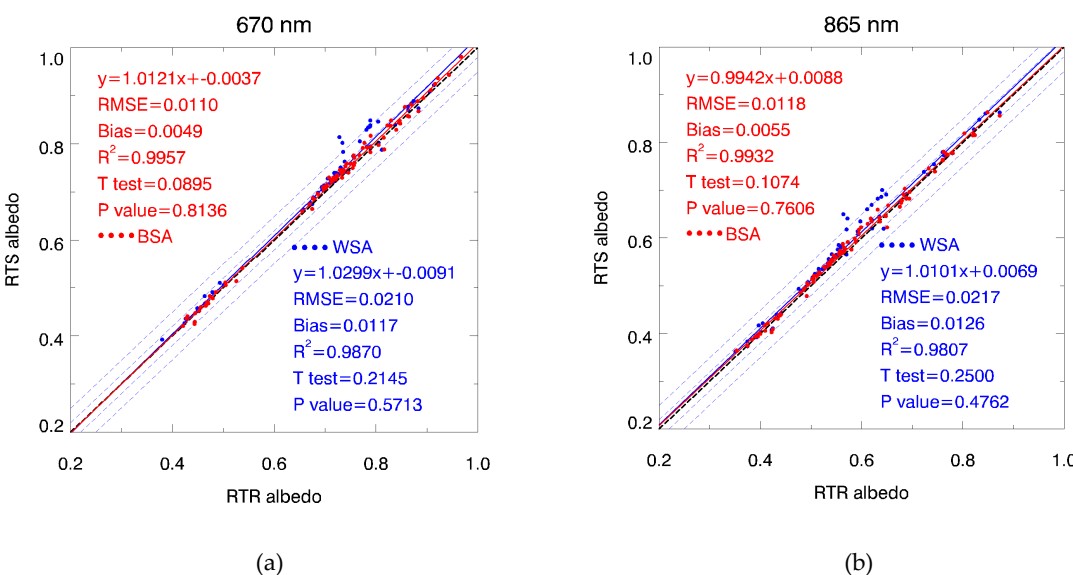

**Figure 10.** The comparison of black-sky albedo (BSA) (red dots) and white-sky-albedo (WSA) (blue dots) retrieved by the two kernel-driven bands in the red (**a**) and near-infrared (NIR) bands (**b**) (i.e., 670 nm and 865 nm). The central dotted lines are 1:1 lines, the outer dotted lines are 0.02 offset lines, and the outermost dotted lines are 0.05 offset lines.

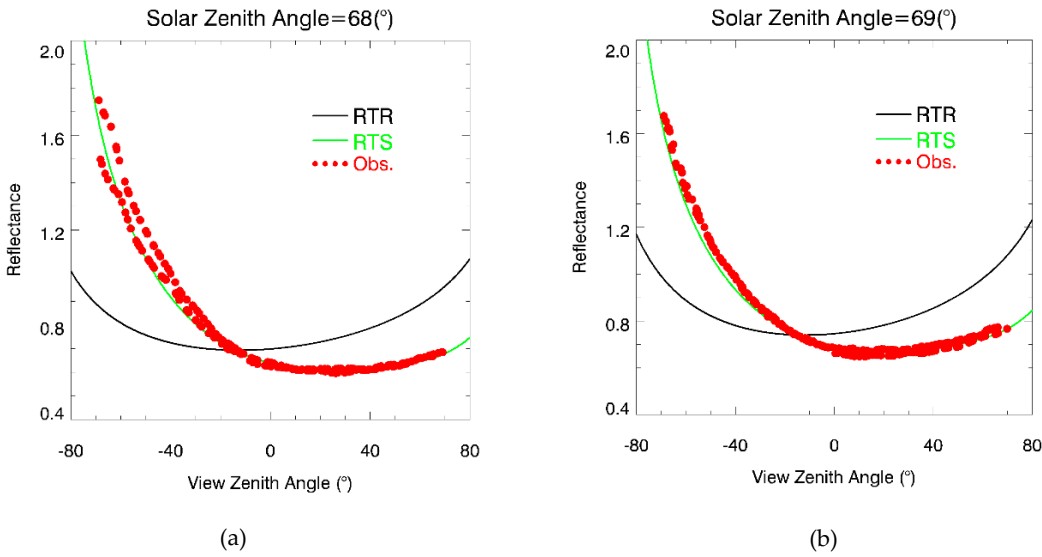

**Figure 11.** The field-measured data (red dots) and reconstructed BRDF shapes using the RTS model (green) and RTR model (black) in the PP and red band (670 nm), where (**a**) and (**b**) represent the cases, and the average SZAs equal 68° and 69°, respectively.

### 4.3. Results of Retrieved Albedo Using POLDER Data

Finally, we compare the difference between these two models in retrieving snow albedo using the POLDER BRF data. The ability of the improved snow BRDF model in fitting POLDER data has been completely investigated [32,39]. In this section, we first compare the difference between these two models in estimating BSA and WSA as a function of wavelength. The results are shown in Figure 12. In general, the retrieved albedos by these two models have good agreement ($R^2 > 0.9$), and the RMSE values are less than approximately 0.02 for all bands. However, the difference in retrieved albedos between two models are statistically significant; the estimated albedos by the RTR model are generally small relative to the RTS model results, and their differences increase as a function of the wavelengths, arriving at 1.34% (1.46%) and 1.59% (1.68%) in the BSA (WSA) in the red (670 nm) and NIR (865 nm) bands, respectively. In the 1020 nm band, the differences in the estimated BSAs and WSAs between two models is 0.02, with the relative percentage accounting for ~37%. Figure 12 shows that these two models are more consistent in the BSA retrievals than in the WSA retrievals for the POLDER data.

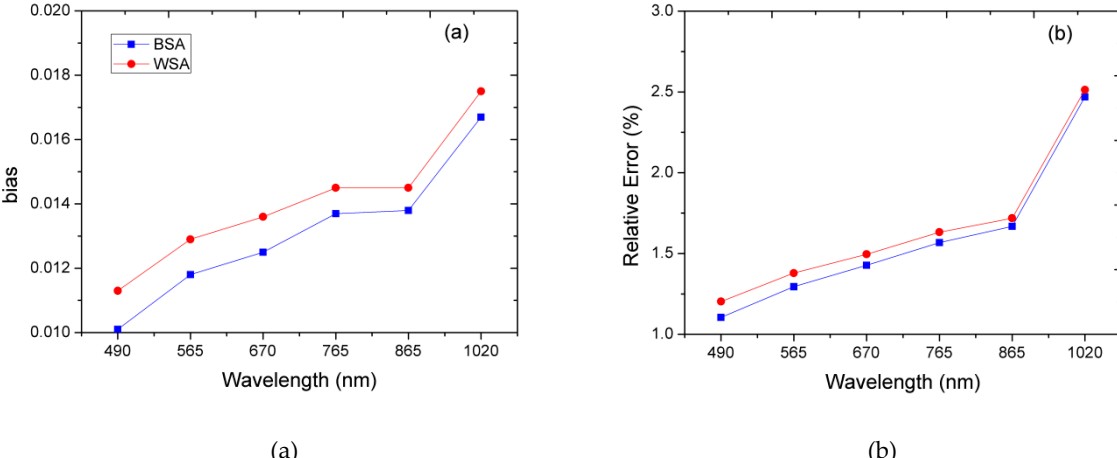

(a)                                                                (b)

**Figure 12.** The comparison of BSA (blue) and WSA (red) retrieved by these two kernel-driven models in 6 spectral channels (i.e., 490, 565, 670, 765, 865 and 1020 nm). Where (**a**) and (**b**) represent the bias and relative error (RE) of snow albedo retrieved by these two models as a function of the wavelength. We further compare the difference between these two models for the POLDER shortwave albedos derived by using Equation (1). The results are shown in Figure 13 and are generally consistent with the results of the narrowband albedo. The shortwave albedo retrieved by the two models also has a high correlation ($R^2 = $ ~0.95), especially in the BSA. However, the shortwave albedo retrieved by the RTR model shows a significant difference relative to the RTS model (P < 0.05), with their differences, which are 1.43% and 1.54%, respectively. Some points are beyond the range of ±0.02, accounting for 17.01% and 15.10% in the BSA and WSA, respectively. This difference is most probably attributed to their ability in fitting POLDER multiangle observations because the potential uncertainties resulting from various factors are completely consistent between the two models.

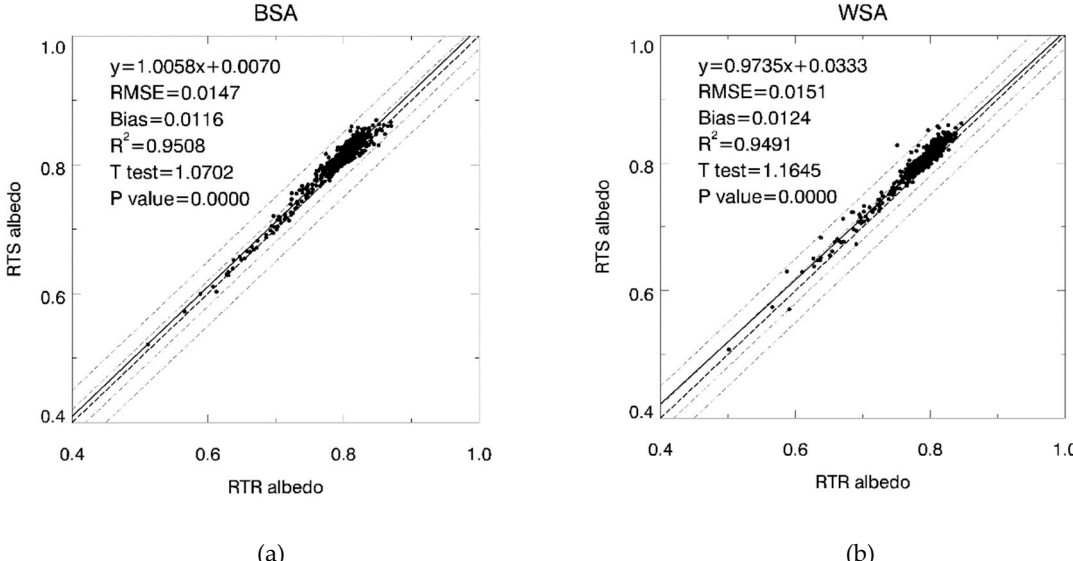

**Figure 13.** The comparison of the broadband BSA (**a**) and WSA (**b**) retrieved by these two kernel-driven models. The thick dashed lines are 1:1 lines, the thin dashed lines are 0.02 and 0.05 offset lines deviating from the 1:1 lines, respectively.

## 5. Discussion

In this study, we comprehensively evaluate the performance of these two kernel-driven models in estimating snow albedo using various data sources. We use the RossThick-Roujean model as a framework for incorporating the snow kernel. The reason is mainly due to the fact that the Roujean kernel exhibits a neglectable hotspot effect and presents more prominent dome-shaped BRDF curves [31]. Such a BRDF curve is generally consistent with the simulation result derived by the bic-PT model, particularly at a small SZA (e.g., SZA < 40°), although the small SZA (e.g., nadir sun) rarely occurs for a natural snow surface. The physics behind this BRDF feature is because the Roujean kernel assumed a surface placed on a flat horizontal plane, which contained a low density of vertical opaque protrusions with low transmittance. This assumption does not consider the mutual shading effects of the shadows of these opaque protrusions. Surprisingly, the results indicate that such an assumption for the Roujean kernel characterizes the BRDF curves of the snow surface very well compared with the LiSparseR kernel used in the operational MODIS BRDF/Albedo model (i.e., RossThick-LiSparseReciprocal, RTLSR), especially at a small SZA (SZA < 40°). In addition, our investigation indicates that the RTR and RTLSR models had similar representations in characterizing the BRDF signatures of the snow, since the geometric optical kernels are basically ineffective (i.e., GO model parameters approach to zero) at a large SZA (SZA > 40°). These results are not provided in this paper due to space limitations. In addition, we just use the snow data with SZAs ≤ 70° and VZAs ≤ 70°, since the quality of snow BRDF data is much better for the simulation data of the bic-PT model, field-measured data and satellite data. For the operational MODIS BRDF/Albedo product, the RTLSR model is not recommended for use in a large solar geometry, which is usually assigned as a lower quality flag. Please refer to the MODIS validation status statement (https://landval.gsfc.nasa.gov/ProductStatus.php?ProductID=MOD43) [32]. Although the snow kernel in the kernel-driven model framework is able to well characterize snow bidirectional signatures at large angles, we just use snow data with SZAs ≤ 70° and VZAs ≤ 70° in this paper to make the models being explored comparable in the assessment.

The snow data also have some potential uncertainties that need to be further discussed. The simulated data show a small sudden break in the PP, which may be caused by the interpolation programme of the bic-PT model [32,39]. However, this small and sudden break in the simulated data should have little effect on the assessment of these two models because the simulated data are generally appropriate for simulating BRDF signatures and snow surface albedo. We just compare the

differences between these two models in retrieving BSA and WSA for the field-measured and POLDER data. Since actual albedos are simply the weighted sum of the BSA and WSA in terms of Equation (5), we expect similar conclusions in assessing the difference between the two models in their estimates of actual albedo. The main difficulty in such an assessment results from a lack of in situ simultaneously observed albedos for the multiangle measurements being explored. In particular, the assessment of the spatial representativeness for the coarse POLDER pixels with a 6 × 7 km spatial resolution is very challenging for the albedo measurements. However, considering that potential uncertainties mainly resulting from multiangle data are completely consistent for the two models, such an assessment does not necessarily require in situ field albedo measurements as a judgement standard because our aim is mainly to assess the ability of the two models in albedo retrievals.

## 6. Conclusions

The kernel-driven BRDF models were originally developed from simplified scenarios of continuous and discrete vegetation canopies. To further improve their abilities to model the anisotropic reflectance of pure snow surfaces, Jiao et al. proposed an improved method by deriving the snow kernel from the ART model [32], which was further used in the kernel-driven model framework. In this study, we aim to assess the performance of this improved method, together with the original kernel-driven model in estimating snow albedo using various data sources of snow. The following findings are highlighted in the results:

(1) In general, the RTR model cannot easily fit snow BRDF multiangle measurements with high accuracy, especially at SZAs ≥ 40°, for which it tends to underestimate (overestimate) reflectance in the forward (backward) direction for various data sources. The overall $R^2$ values derived by the RTR model have a range of 0.21–0.76. In contrast, the RTS model performs very well in capturing snow BRDF measurements and presents a high correlation ($R^2$ = ~0.9) in fitting various snow BRDF data, which indicates that this snow kernel in the kernel-driven BRDF model presents an improved capability in capturing snow scatterings. This finding is generally consistent with the conclusions derived in a previous paper [32].

(2) Although the intrinsic albedos (i.e., BSA and WSA) with these two models are highly correlated, a significant difference has been examined between them, This difference is somewhat dependent on the band, SZA range and BRDF sampling based on the simulation and observation data being explored. The bic-PT model is able to simulate comprehensive BRDF data sets and albedos as a judgement standard. The assessments using these simulation data show that the snow albedo retrieved by the RTR model presents a statistically significant difference. At SZAs = 30°–70°, the albedos retrieved by the RTR model are underestimated by 0.71% and 0.69% in the red and NIR bands, respectively. In contrast, the RTS model shows high accuracy with the simulation albedos over all SZAs. In general, the retrieved albedos by the RTS model at SZAs = 30°–70° have a negligible bias and do not present a significant difference. Along the azimuthal sampling planes from the PP to the CPP, the RTS model presents a high accuracy and stability relative to the RTR model in retrieving albedo. However, in the exact CPP where the BRDF variations are very weak, the findings of the two models are not significantly different.

(3) The assessments with the multiangle observations of both the field measurements and satellite data further confirm that there is generally a significant difference in the retrieved intrinsic albedos by these two models. Specifically, the albedo retrieved by the RTR model shows a significant underestimation compared with the RTS model results, especially for the WSA, somewhat depending on different SZAs. For the field snow measurements, the difference in albedo retrievals between the RTR and RTS models is not prominently significant. However, for the entirety of the POLDER data, in which a large range of SZAs is captured on a global scale, the difference in the albedo retrievals between these models is significant, with the RTR model underestimating the values by 1.43% and 1.54% in the BSA and WSA, respectively, compared with the RTS model.

In summary, this work presents an improved method using multiangular remote sensing techniques and demonstrates the availability of these models using a kernel-driven model framework to

achieve improved reflectance retrievals of snow multiangle data and, hence, improved the subsequent retrievals of intrinsic albedos.

**Author Contributions:** Conceptualization, A.D. and Z.J.; Data curation, A.D., J.I.P., J.G., S.Y. and L.C.; Formal analysis, A.D., Z.J., Y.D. and S.Y.; Funding acquisition, Z.J.; Investigation, X.Z., J.G. and R.X.; Methodology, Y.C.; Validation, L.C.; Writing—original draft, A.D.; Writing—review & editing, A.D., Z.J., Y.D., X.Z., J.I.P. and L.M.

**Acknowledgments:** This work was supported by the National Key R&D Program of China (2018YFA0605503) and the National Natural Science Foundation of China (41571326 and 41801237). The work of Linlu Mei is supported by the SFB/TR 172 "ArctiC Amplification: Climate Relevant Atmospheric and SurfaCe Processes, and Feedback Mechanisms (AC)3" funded by the German Research Foundation (DFG, Deutsche Forschungsgemeinschaft). The POLDER-3/PARASOL BRDF databases were developed by the Laboratoire des Sciences du Climat et de l'Environnement (LSCE) and provided by the POSTEL Service Center.

**Conflicts of Interest:** The authors declare no conflicts of interest.

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
