# Peer review of "Evaluation of the Snow Albedo Retrieved from the Snow Kernel Improved the Ross-Roujean BRDF Model"

_remotesensing, doi:10.3390/rs11131611_

Round 1

Reviewer 1 Report

Well done!

You have a mismatch between "a" and "alpha" in eq. (1) and the description afterward, and "WAS" and "WSA" in eq. (4) and the paragraph before it.

Author Response

Dear Reviewers,
Thanks very much for your positive and valuable suggestions on this manuscript. Those suggestions helped improve the quality of this manuscript. We read these suggestions carefully and made corresponding changes.

Throughout our detailed reply below, our point-to-point responses to the comments are written in red. Please note that a "track changes" (highlighted) version of our revision is provided, which we hope facilitates the re-review process.

We hope that our revised paper addresses your concerns and that you will now find the manuscript acceptable for publication in Remote Sensing.

Please note that these resulting revisions did not change the major findings of the paper.

Thank you and best regards.
Sincerely,
M.S. Anxin Ding & co-authors

Response to Reviewer 1 Comments

Well done!

Thanks very much for the endorsement of this paper.

Point 1: You have a mismatch between "a" and "alpha" in eq. (1) and the description afterward, and "WAS" and "WSA" in eq. (4) and the paragraph before it.

Response 1: Thanks very much for this careful check.

Sorry for these typos. We have revised “a” to “alpha”, please see the Eq. (1) and others. We also have revised “WAS” to “WSA”. Please see lines 183 and 185, in the revised manuscript.

Reviewer 2 Report

I am pleased to read this manuscript about snow BRDF kernel evaluation. The deficiency of MODIS kernel-driven model in simulating snow surface BRDF is a key issue influencing the current BRDF/albedo products quality. This topic is of interest to readers; however, the motivation of this paper is not clear. I cannot see the specific meaning beyond the cited Jiao's paper. Suggest the authors analyze insightful and make it clearer. Besides, the expression and format need careful proofreading. I would recommend major revision before considering to be published.

1. The purpose of this paper is not clear. 1) Line 105~108 suggests it is to check the improvement of this snow kernel based on Jiao et al. [32]. Why not using the current MODIS operational kernels? 2) Compared to the evaluations of the improvements brought by adding the snow kernel in Jiao's paper [32], is there any improvement in the methods of this manuscript?

2. Table 1, Would the authors explain why only choose red and NR two bands? Besides, why exclude SZA>70? (continue in the following comments)

3. L146-147. “In this study, we used pure snow 147 data with high quality and further screened out the datasets with SZAs > 70° and VZAs > 70°. “ Snow surface mainly happens in high-latitude regions, where the SZA can exceeds 70 in many seasons. Not assessing the kernel performance when SZA>70 should be noted as a limitation.

4. L233, the paper did not simulate the VZA>70, so under such case, the performance of RTR and RTS can not be evaluated. Only comparing them seems not providing useful information.

Minor

Line 52-54, Rephrase the sentence“Snow albedo, defined as the ratio of upwelling radiant energy to downwelling irradiance… “. The albedo definition is not only for snow surface. Besides, it would be better to use simple sentences here rather than repeating the same idea in a clause.

Line 61, who is Barry?

Line 98, I cannot understand this sentence. What is the relationship between the citation and the RPV model?

Line 109, the font of the paragraph is not consistent with previous ones.

Round 2

Reviewer 2 Report

Thanks much for the detailed response with summary of the answers and revisions. All my questions and comments have been addressed. I agree to accept the publication in present form.